# Alternative Food Networks: Perceptions in Short Food Supply Chains in Spain

Sttefanie Yenitza Escobar-López [1], Santiago Amaya-Corchuelo [2,*] and Angélica Espinoza-Ortega [1,*]

1   Instituto de Ciencias Agropecuarias y Rurales (ICAR), Universidad Autónoma del Estado de México, 50000 Toluca de Lerdo, Mexico; osiris_1018@hotmail.com
2   Campus de Jerez, Universidad de Cádiz, 11406 Jerez de la Frontera, Cádiz, Spain
*   Correspondence: santiago.amaya@uca.es (S.A.-C.); angelica.cihuatl@gmail.com (A.E.-O.)

**Abstract:** Alternative spaces for the consumption of non-conventional foods as short commercialization chains have been increased worldwide, as well as the interest in understanding the dynamics developed in those initiatives from a social approach, has increased. This work aimed to analyze the Organizers, Producers and Consumers' perceptions that participate in short food supply chains in ecological markets in the south of Spain. The Conventions Theory framework was used by applying a quantitative and qualitative methodology. A total of 159 questionnaires were applied (three to Organizers, 15 to Producers and 177 to Consumers). The questionnaire considered items related to social conventions for seven worlds (Domestic, Civic, Market, Industrial, Opinion and Inspired) and sociodemographic information. Kruskal-Wallis and Mann-Whitney tests were used to analyze the information. Results show that coincidences and divergences are observed in the importance given to the worlds; the Civic, Inspired and Opinion worlds are perceived similarly by the three types of actors and are related to the interest in how these activities benefit the environment. Differences are attributed to the role played by each type of actor and to socioeconomic aspects. Results can materialize into different strategies to improve these initiatives and reach more consumers.

**Keywords:** alternative spaces; short chains; ecological markets; nutritional trends; perceptions

## 1. Introduction

Human activity has had negative effects on the planet. The effects have been so powerful that there has been a period change known as Anthropocene [1]. Agriculture has substantially contributed to these changes [2] with a globalized and industrialized agri-food system that, apart from the environmental problems, has also had an effect on the people's diet and health [3,4].

As a consequence of this situation, several sectors in society have emerged with a negative perception about the agri-food system and are interested in creating a socially, economically and environmentally fairer world [5]. These concerns have increased in recent decades. More and more people are worried about the way in which their consumption and actions can affect the environment [6]. In this way, consumer groups have emerged with ethical values characterized by a concern about what they eat [7] and their constant search for green and healthy products with specific quality and authenticity features [8,9]. These reflections were the origin for the development of short food supply chains (SFSC) [10].

## 2. Literature Review

### 2.1. Short Food Supply Chains

These initiatives can potentially improve and foster agricultural production, rural development, land use and local economic development and consumption [9,11]. These alternative spaces happen within the dominating economic and political structure, but with an ideology and a set of practices that belong to different economic logics [5]. Furthermore,

they represent a connection between the city and the rural environment and also a way to establish new links between agriculture and society (producers and consumers) via the direct contact between them and with the foods [12,13].

These channels have been defined as the interrelationships that are established among different agents of the agri-food chain, agents directly involved in the production, transformation, distribution and consumption of new foods [14]. Therefore, these agents also have the ability to create a new socialization and space for food through the knowledge, experience and closeness [15], in a contemporary and dynamic process [16] in which there is a general trust [10,17].

The issue of consumer or customer satisfaction is a relevant and current research topic at all commercial levels, as has been shown for large firms where high customer satisfaction can diminish the negative effects of corporate social irresponsibility [18,19]. In this sense, standard regulations along food supply chains may positively improve each stage of the supply chain [20].

In the broad context of short food supply chains, there are specific proposals for new spaces for production, commercialization and consumption of differentiated foods [18]. These are the so-called Alternative Food Networks (AFNs) [21,22]. Some examples of this type of networks are, community agriculture, box schemes, fair trade [23,24], farmers' markets [25] and ecological markets, among others.

Ecological or organic foods can be found among the offered products. The meaning of these concepts changes depending on the country of production. Independent of nomenclature, the main value of these products comes from their links to an alternative agriculture, in which the productivity of soils and the natural cycles in plants and animal production are maintained. This ideology is constructed in contrast to the use of inputs with adverse effects, such as synthetic fertilizers, pesticides, growth regulators and additives in animal feeds [26–28].

Although the organic sector is still relatively small, the demand for organic food is increasing around the world [29], mostly in European countries [30,31], but there are differences depending on the country. In this context, Spain is the main producer and supplier of organic products, but just 25% is destined for domestic consumption, the rest is exported. Nevertheless, demand for the ecological market keeps growing, increasing 12.55% in 2018 in comparison to 2015. Spain is now among the first ten countries in terms of domestic market volume [32].

Andalusia is the leading Spanish region in regard to the cultivated area and production of ecological foods, with 46% of the total for the country [33,34]. This has an influence on the local consumption and it is estimated that more that 54% of its population has consumed ecological products [35].

Data indicates that short commercialization chains have an important presence in the acquisition of ecological foods and that it is growing [36]. According to the study carried out by the Ministry of Agriculture and Fisheries, Food and Environment in 2017, half of the Spanish consumers of ecological foods obtain them through channels in which the majority of the offered products are ecological. According to the mentioned study, 29% of ecological foods buyers purchase these foods exclusively in specialized channels. The most common ones are direct selling, specialized shops, herbalist shops, small shops or specialized traditional market stalls [37].

These urban spaces have slowly become consolidated as places for the responsible sale and consumption of food [38,39] that provide three types of benefits: on health, well-being through the solidarity to farmers, and in minimizing the ecological footprint of production [40].

In the European context, there are different works which have dealt with AFNs. Some authors who have analyzed these markets determined that their development and growth depend to a large extent on the knowledge of the role of consumers and also on the satisfaction of their needs [29,41–43]. Although the potential importance of individual consumers and communities is noteworthy [44], the consideration of the rest of actors involved in

food chains (in order to correctly understand their signifiers and practices) becomes of relevance. However, the main gap of those studies is that production, consumption and retail channels [16,29,40,45–48] and organizers [25] have been analyzed separately. Therefore, to obtain a global vision of AFNs, behaviors, motivations and perceptions of all involved actors should be taken into account [4]. In this manner, a solid alternative can be constructed to face the predominant ways for the distribution of foods [49] and to strengthen these proposals. This study proposes the following hypothesis:

**Hypothesis 1 (H1).** *The global perspective of AFNs can be determined if the perceptions of all the types of actors that comprise them are analyzed.*

The lack of studies may be due to the absence of methodological tools that allow analyzing the various actors involved under similar bases, regardless of their role in the AFNs.

*2.2. Theoretical Framework*

The Conventions Theory (CT) is recommended [50] for the study of the perspective in the production and consumption chains of ecological products [51], as an analytical tool that allows for the combination of a wide range of analysis parameters from a holistic perspective.

The CT originated in France to assess mainly agri-food economic aspects. Boltanski and Thévenot, in the 1990s, in their work *De la justification. Les économies de la grandeur* laid the foundations for the sociological approach of the CT [51]. The methodology is based on articulating six groups of variables called *worlds of justification* or action assessment (tools to analytically measure the actors' dynamics and their assessments) [52]. "Conventions" was the name given to these routine practices, agreements or justifications, which are part of the *Inspired, Industrial, Market, Domestic, Opinion* and *Civic* worlds (established by Boltanski and Thévenot) [53].

In any form of coordination in the political, economic and social sphere, agreements (in constant strategic negotiation) are required, based on the construction of common perceptions about the context and the interaction [53,54]. Therefore, the CT is a holistic approach that allows for the analysis of the behavior of actors in daily actions inside a group, where subjective agreements, assessments and different types of perceptions are developed [55,56].

Conventions are defined as a system of reciprocal expectations in the behavior of others [54] that may be analyzed [57] since these are centered in scoring, judging, justifying and criticizing forms on how links are established between cognitive, moral and material concerns [43].

In that sense, people are assessed through their action [52,58] in different worlds of justification, stating that actions are legitimized by support on particular view points of the common good [57].

Therefore, TC enables the better understanding what regulates actions from different actors and how they perceive their actions [55], that is, ways of valorization to face criticism and justification [57].

These worlds or interaction ways among the actors of food chains are presented by means of problematic relationships, and at the same time, they complement each other with some specific limits and agreements, which contribute to the coherence and maintenance of any of these networks [59,60].

In this sense, taking into account that participants in AFNs can stay financially, politically and socially active [61], it is relevant to identify whether there is an appropriation of the goals by all the actors participating in them. It is also relevant to identify the coincidences and differences in the dynamics of their actions.

There is a gap in the knowledge on a methodological approach that considers all of the different actors involved in AFNs. This is why the CT approach can contribute to their knowledge. In that sense:

**Hypothesis 2 (H2).** *The CT is a useful approach to analyze the perception that the different actors involved on AFNs, have about these spaces.*

**Hypothesis 3 (H3).** *There are coincidences of objectives and perceptions of the different types of actors involved in the AFNs.*

Therefore, the objective of this work was to analyze the perceptions of the social actors (organizers, producers and consumers) participating in ecological markets in the south of Spain, based on the social conventions approach.

## 3. Materials and Methods

Pre-fieldwork was first carried out to identify ecological markets in the Cádiz area. An agro-ecological network and a cooperative society were contacted. The first one has the objective of defending and promoting agroecology (not just ecological foods). They are focused on the management of agro-ecosystems, commercialization systems and human relationships. According to their rules, they search for sustainability in all its dimensions (ecological, social, economic, cultural and political) [62]. They work in the province of Cádiz and the network is formed by people, organizations and businesses. The Andalusian cooperative society focuses on production and consumption of local and traditional ecological products and are ecologically certified by the CAAE (Consejo Andaluz de Agricultura Ecológica (Andalusian Council for Ecological Agriculture)) [63]. Both organizations are the most significant alternative food networks in this territory.

Non-probabilistic convenience sampling was used to select participants following the criteria that people were older than 18 and were interested and willing to participate [64]. A total of 159 persons participated (3 Organizers, 15 Producers and 177 Consumers).

The methodology applied in this research was qualitative and quantitative, with semi-structured interviews and questionnaires. The questionnaire had two sections: (1) a series of items related to social conventions for each world (*Domestic, Civic, Market, Industrial, Opinion and Inspired*), items that were adapted according to multiple works [46,50,51,55,57,65] (Table 1) that were rated with a 5-point Likert scale (1 = Total disagreement and 5 = Total agreement); and (2) sociodemographic information (age, gender, family income, educational level and marital status).

**Table 1.** Conventions assessed for each World.

| World | Conventions (Items) |
|---|---|
| *Market* | Prices for ecological products are appropriate. <br> The ecological products are worth the extra cost. <br> The quality of ecological foods is higher in comparison to conventional ones. <br> Quantity and variety of ecological products in these spaces covers the consumers' demands. <br> These spaces guarantee the sale of quality ecological foods. |
| *Industrial* | All those participating in the production and sale of ecological foods are committed to their work. <br> There is a control process in the production and sale of ecological foods. <br> The number of producers and ecological products in these spaces is appropriate. <br> Products offered in these spaces require some kind of certification about being ecological. <br> The organization of the market is adequate. |
| *Civic* | Agrochemicals should not be used in the production of ecological products. <br> Animal welfare should be taken into account in the production of ecological products. <br> Apart from producers and consumers, society in general benefits from these spaces. <br> These spaces foster relationships with equality, respect and equity. <br> Ecological foods should be accessible to all. |
| *Domestic* | These spaces allow for the direct contact with organizers, producers and consumers. <br> This type of establishment and the offered ecological foods provide benefits for those involved. <br> There is a guarantee that the sold products are ecological. <br> My relatives are actively involved in the purchasing of ecological foods. <br> Production, sales and consumption of ecological foods must be a lifestyle. |
| *Inspired* | Boredom, amusement, kindness, guilt, satisfaction, displeasure, sadness, wish, happiness, nostalgia, joy, pride, disappointment, anger, tranquility, shame, concern, pleasure. |
| *Opinion* | These spaces and the consumption of ecological foods are in fashion. <br> The quality of ecological foods must be backed by a trademark. <br> The certification used in these spaces is enough to endorse ecological foods. <br> The promotion of these spaces and also of ecological foods is essential. <br> In these markets, the consumption of ecological foods must be recommended. |

*Analysis of the Information*

A database was created in Excel and worked independently by type of actor (producer/organizer/consumer). The scores for the conventions of each world were added and averaged for each participant. These values were used to calculate the median and interquartile range for each world since Likert scales are non-parametric data [66]. The medians and interquartile range were used in the nonparametric Kruskal-Wallis and U-Mann-Whitney tests [66], to identify significant statistical differences ($p < 0.05$) in the perception of the worlds per type of actor (Table 2). An amoeba graph was made to have a visual comparison of the worlds' values per type of actor (Figure 1).

**Table 2.** Comparative analysis of the actors according to the present worlds.

| Worlds | Organizers n = 3 | | Producers n = 15 | | Consumers n = 177 | | *p* |
|---|---|---|---|---|---|---|---|
| | $\tilde{X}$ | IQR | $\tilde{X}$ | IQR | $\tilde{X}$ | IQR | |
| *Inspired* | 3.3 a | 0.9 | 3.2 a | 1.4 | 3.1 a | 1.7 | 0.990 |
| *Civic* | 4.6 a | 0.6 | 4.6 a | 0.8 | 4.6 a | 1.8 | 0.714 |
| *Opinion* | 4.2 a | 0.8 | 4.2 a | 1.2 | 4.0 a | 2.4 | 0.096 |
| *Industrial* | 4.0 a | 0.4 | 4.6 b | 1.0 | 3.8 a | 2.2 | 0.000 |
| *Domestic* | 4.4 a | 0.6 | 5.0 b | 0.4 | 4.0 a | 1.6 | 0.000 |
| *Market* | 4.2 ab | 0.6 | 4.2 b | 1.2 | 3.8 a | 2.0 | 0.002 |

IQR = interquartile range; *p* = value of the Kruskal–Wallis test ($p < 0.05$); a, b Mann-Whitney U test by rows ($p < 0.05$).

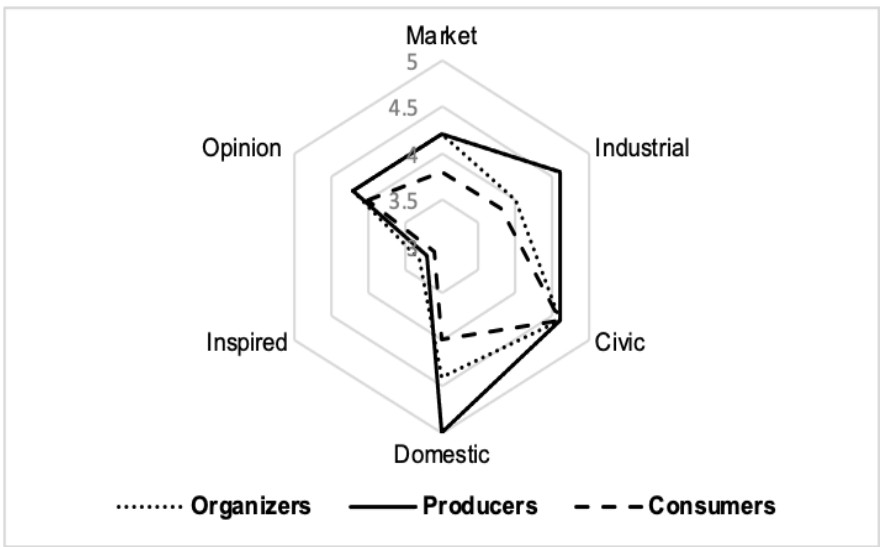

**Figure 1.** Importance of the worlds according to the type of actor.

Sociodemographic information was analyzed with descriptive statistics. The age variable was classified into sociological generations: Millennials (1983 to 2002), Generation X (1965 to 1983), and Baby boomers (1943 to 1964) [67].

Finally, the statements of 20 interviewed people were used to complement the analyzed information about the study's social practices [68].

## 4. Results

Two aspects were observed initially when analyzing results. Firstly, virtually all worlds received high scores from the three types of actors, especially the *Civic* and *Domestic* ones (Figure 1). On the contrary, the *Inspired* world received the lowest score from the three groups. The second aspect to highlight was that there were differences in the actors' valuation about the assessed worlds (Table 2). They were statistically significant for the *Market*, *Industrial* and *Domestic* worlds ($p < 0.05$), but not for the *Civic, Inspired* and *Opinion* ones ($p > 0.05$).

Results showed that groups were also different in relation to the socioeconomic variables of gender, generation, educational level and marital status (Table 3).

**Table 3.** Sociodemographic characteristics of the participating actors.

| | Variable | Organizers (n = 03) | Producers (n = 15) | Consumers (n = 177) |
|---|---|---|---|---|
| Gender | Male | 0 | 53.3 | 39.5 |
| | Female | 100 | 46.7 | 60.5 |
| Sociological generation | Millennials (aged 18–36) | 33.3 | 13.3 | 42.4 |
| | Generation X (aged 37–57) | 66.7 | 86.7 | 34.5 |
| | Baby boomers (aged 58–75) | 0 | 13.3 | 23.2 |
| Educational level | Basic/Preuniversity | 0 | 53.3 | 23.7 |
| | University/Postgraduate | 100 | 46.7 | 76.3 |
| Marital status | Single | 66.7 | 20.0 | 40.1 |
| | Married | 33.3 | 73.3 | 50.8 |
| | Other | 0 | 6.7 | 9.1 |

The level of importance or connection that each of the actors gave to the different worlds was related to the role they play in the markets. Those nuances generate contrasting stances or perceptions, specific for the assessed markets (depending on the type of actor).

Organizers and Consumers have similar perspectives about the worlds and the only difference is that Organizers consider that the *Market* world is more important. On the contrary, Producers were different, especially because they placed more importance on the *Market*, *Industrial* and *Domestic* worlds. The characteristics of the groups of actors can be summarized as it follows:

*Producers.* Men and women, married in most cases, with a wide variety of educational levels (from basic education to university). The most outstanding characteristic was that they all belonged mostly to Generation X (Table 3). They gave higher values to practically all variables, being the *Domestic* world the most important one, followed by the *Industrial* and *Civic* ones, and with less significance given to the *Opinion* and *Market* worlds. *Inspired* world is irrelevant to them. (Figure 1).

*Consumers.* People with high educational level, mainly young women from the Millennial generation and, to a lesser extent, Generation X (Table 3). Surprisingly, they had the lower scores for all the worlds. They stand out from other groups because the *Market* and *Industrial* worlds are irrelevant to them. As with other groups, the *Inspired* world is also unimportant. The most important world for them is the *Civic* one, followed by the *Opinion, Domestic* and *Industrial* ones (Figure 1).

*Organizers.* The sample was formed just by women with high educational level, mostly single and from Generation X (Table 3). This group presents intermediate values in all variables with respect to the other groups. The *Civic* world is the most important one for this group, then the *Opinion* and *Domestic* worlds, followed by the *Industrial* and *Market* ones. The *Inspired* world is also irrelevant to them (Figure 1).

## 5. Discussion

Although the Conventions Theory does not assign a hierarchical value to the worlds [59,69], it determines that they have controversial relations in which they try to reduce the importance of the other ones, but as there can be multiple action justifications operating at the same time, the worlds end up complementing each other and are found in hybrid forms [59,60,69–71]. It is in that diversity that coincidences and divergences are observed in the importance given to the worlds by the actors in those spaces.

### 5.1. Coinciding Worlds

As it was already mentioned, the three types of actors perceive the *Civic, Inspired* and *Opinion* worlds in a similar way.

*Inspired world.* In this world, justifications emerge from the immediate relation of people with an external source from which all possible values flow. It is independent from others' acknowledgement and it emerges through emotions. The expressions are varied and multifaceted: saintliness, creativity, artistic sensitivity, imagination, emotiveness, etc. [57].

This world has not been taken into account in the study of the systems [43]. However, it cannot be left out to focus on the economic aspects, as it has traditionally been done in other studies [72]. Its application is recommended to score products of the land that try to surprise consumers with creative innovations [73] in which the emotional factor is an essential one.

It is proposed that in the study of social movements, the affective or emotional dimension should be considered [47]. Therefore, it represents a window of opportunity for a deeper study, as it was the less valued world by the three types of actors. However, Consumers' statements indicate that visiting ecological markets generates in them positive emotions (such as amusement, kindness, happiness, tranquility) because of carrying out an activity with multiple benefits:

> "*I am happy here because of what I eat, knowing who produces it and the way it is produced...*" (Consumer, age 35).

Visiting these spaces made Consumers feel emotionally better in the face of a civic concern in response to negative emotions such as disappointment or sadness due to the deteriorated environmental situations and with concerns for a better world.

*Civic world.* This is related to contributing to the ecological well-being and also to the health and security of all citizens. It fosters social cohesion, well-being and justice [73], as it prioritizes collectives and gives up on particular interests [43,52,57,73].

The civic aspect is the most important one in these alternative proposals. The three types of actors value the benefits for the environment and for the people participating in the markets, and also the possibility of more people having access to these services and products. Relationships with equality, respect and equity are also much appreciated. This is the statement of one of the Organizers:

> *"These markets were created to foster the consumption of ecological, seasonal and local foods, and at the same time [to strengthen] personal relationships and to offer everybody the chance of consuming them . . . "* (Organizer, age 37).

For some sectors, the *Civic* world must have some elements that guarantee it, and it is here where the *Opinion* world takes action.

*Opinion world.* Also referred to as public or renowned, this world makes reference to trademarks, logotypes, labels or commercial presentations (aspects that allow consumers to easily recognize a product) [57,73,74]. The measuring of these aspects depends on the conventional signs of public esteem, that is individuals offering recognition [73], and not linked to personal self-esteem [43,57]. This world is not based on reality, but on power relations between conflicts of opinion [57].

Although the *Opinion* world was not the most important, the three types of actor value it similarly (Graph 1). When making reference to the fame of products and services, these spaces should be open and should not hide information, so that a positive recognition is generated. This coincides with the results of the current work, where the promotion of these markets is considered essential. The use of a trademark and a certification is also a sign of quality and type of production.

As was mentioned, these ecological markets have an ecological certification from the CAAE (Consejo Andaluz de Agricultura Ecológica (Andalusian Council for Ecological Agriculture)). It is considered an essential requirement that gives added value to the offered products (but not an absolute one) [47]. Apart from the trademark and certifications, in these spaces, the *Opinion* world is based on the trust on the Producers, as can be seen in the following statements:

> *"The certification and the trademark are important for producers and consumers, but it is more important that these markets become known. It is also more important to promote the benefits of coming and knowing how each food is produced..."* (Organizer, age 40). *"We have a certification and it is shown because it is important to have it, but we are not asked about it, people like more talking to us..."* (Producer, age 45). *"The certification is important, but it is more important to know who produces what we eat, that is why we recommend it so much..."* (Consumer, age 38).

Consumers construct different imaginaries about alternative networks. Support to local retailers is important, but it must be based on the trust that the quality is being stated in an honest way [5]. Although the word-of-mouth promotion is essential, these spaces have implemented some additional strategies (related to the *Industrial* world) for the promotion, recognition and presence of these products.

## 5.2. Divergent Worlds

Identifying the coinciding worlds is as important as identifying the divergent ones. Divergent does not necessarily imply a contrast, as it can explain the social interaction as an act of negotiated coordination subject to the justification of strategic positions [51,59].

*Industrial world.* Its value is based on the technical efficiency, the experience, the professional ability and the standards to offer trustworthy and long-lasting products or goods [73,74]. The quality would be assessed with objective and measurable parameters [43], which is guaranteed by standardization [70].

Taking into account the above mentioned, it is logical that the Producers obtained the highest values in the *Industrial* world and that their perception is different from the one of Organizers and Consumers. They are the ones that can measure the technical efficiency and the standards to offer reliable products and services (Table 2), and that can be seen in the statements below:

*"Every day we work so that the products we sell are good and with quality, we have a good production control so that consumers are satisfied with what they choose..."* (Producer, ager 55).

In this way, Producers establish the commitment with their work via the control of the production, the sales and the availability of their products, as well as an adequate organization of the markets. This coincides with results from Finland and the UK where alternative retailers from the showed a higher affinity with this world [46].

Furthermore, it is established that technical efficiency is linked to a common good in time and space, that is, through "progress" and "update" [70], and it is maintained with organizational tools used for planning and for future investments [52,57]. These spaces have a constant promotion campaign in social networks, where information can be obtained about the schedules and timetables, and also about the offered products (these changes depending on the season and availability from each producer). This foray into marketing via the digital platforms has been of great help during the COVID-19 pandemic period. A system for the placement of orders via telephone/website was implemented for home deliveries (by Organizers) or for Consumers to collect the products in one of the distribution points.

*Domestic world.* In this world, the value assigned by people depends on a trust hierarchy based on a chain of personal dependencies. The person is not separated from belonging to a family, a legacy and a past. The place in this network of dependencies (from where the person extracts his/her own authority) should be known [52,57,73].

For the Producers, the *Domestic* world received the highest value, and this is another aspect that is a difference in relation to the other actors. Although the three groups considered the direct contact with the different participants in the markets to be important, personal contact is perceived by producers as the guarantee of the offered type of products (ecological and local, among others) in these spaces. Trust relationships, traditionality, family and proximity are highly valued in this world. It is considered relevant that relatives get involved. This was expressed by the following statements:

*"My wife helps me and we all come to sell. My young kids like it. People get to know us, they ask us questions about how products are made and sometimes they visit our garden..."* (Producer, age 45). *"My family likes coming with me to buy these products. We do not just buy products. More things can be done here, we live close by and they are local products..."* (Consumer, age 50). *" ... to know who the producer is and the way in which foods are produced gives us peace of mind, that can only be done in these markets where you can talk to the producers ... "* (Consumer, ager 35).

It is clear that the local origin of these products is highly appreciated and helps to establish trust relationships with Producers. Socialization and bonding boost the participation of the different members, helping in the development of links and justifying their presence [47]. These are essential aspects in the short chains for the commercialization of differentiated foods, as they favor the generation of meaning and extra value for the products [75].

*Market world.* In this world, the important actors are consumers and sellers. The first ones are perceived as ideal when they have resources. Their virtues are locating and taking advantage of the market opportunities, being free from any personal link and staying emotionally under control [57]. For the sellers, the quality of the products is assessed with the principles that control the market, where price is the main criterion [73]. This world is focused on competitiveness and profit for producers, and on satisfaction and usefulness for consumers [43,73].

It is clear why the three types of actors have different perceptions about the *Market* world. Producers and Organizers gave more importance to this world and they were the ones most interested in an adequate operation of these spaces and their success. For Producers, the presence in these markets guarantees the sale of quality ecological foods that require more work and production time. They consider that these features make these foods better than conventional ones, so they have a higher price:

> *"It is important that everything works adequately. Several days before the market is held, we meet with the producers, organize all activities and verify the goods that will be sold . . . "* (Organizer, age 37).

The statements contribute to what has been determined by the literature, which establishes a relationship between the *Market* world and the competitiveness and profit of companies and also the usefulness of products and services. However, consumers and their satisfaction are also mentioned [18–20,43,46,73,74,76]. It is relevant that this world did not obtain high scores from consumers.

As already mentioned, the grade of importance each type of actor considered for the different worlds is related to the role each of them plays in the markets. The socioeconomic characteristics of each type of actor also have a strong influence.

### 5.3. Influence of the Socioeconomic Factors

Several works suggest that gender, age, occupation and education (among others) have an influence on the perceptions and general attitudes of individuals towards foods and sales establishments [77].

Regarding gender, more than 60% of the interviewees in this study were women. This coincides with what other authors have established about these spaces being feminized [13]. Several works have reported that women have more pro-environment attitudes and take a more active part in this type of initiatives, being described as "female citizen consumers" [13,45,78]. For example, Danish women (in their consumption of differentiated foods) showed a higher affinity with the *Civic*, *Domestic* and *Market* worlds [79]. In the current work, groups with a higher proportion of women were the Organizers and the Consumers. Nevertheless, the *Civic* world had the same importance for the three types of actors. The higher proportion of men in the group of Producers explains the importance attributed to the *Industrial* and *Market* worlds.

In relation to the educational level, it has been identified that the *Civic*, *Industrial*, *Inspired* and *Market* worlds are related to a high educational level [79]. In the current work, the educational level was high for the three type of actors, although lower for the Producers (over 50% only have secondary education with no university education). Different studies have demonstrated that consumers with a high educational level attend this type of initiatives [45,80]. Some works have established a link between the higher educational level and the possibility of choosing alternative foods due to the information available [8,81].

To this respect, there is the importance of "*cultural intermediaries,*" understood as those elements that favor the incorporation of new cultural and consumption aspects [82]. In this sense, the best cultural intermediary is education, which has increased in the younger generations and is a determining factor in the characteristics of sociological generations.

In the current work, the highest proportion of Producers belonged to Generation X (86.7%). They are considered to be a conservative generation that values family, with no dependence from institutions and appreciating leadership [83,84], therefore the affinity with the *Domestic*, *Industrial* and *Market* worlds. The affinity of Organizers with the *Market* world also coincides with a higher proportion of people from Generation X. On the contrary, the Millennial generation is more present in the Consumers and is described as hedonic and skeptical, but at the same time interested in an ecological life with civic mentality, open to new products or services [77,83,84]. Young generations are more alternative and consider that they can change the world through their consumption. Other generations take advantage of this situation.

*5.4. Actors' Stances*

In Italy, a study on short chains established that consumers and their fair-trade consumption habits influence and foster new marketing strategies (in farmers) that are more ethical and related to trust [85]. This coincides with the results of the current work. The origin of everything is the Consumers' *Civic* world, valued by other actors but strengthened with the worlds that are important to them depending on the situation. For Producers, the *Industrial* and *Market* worlds, and the *Domestic* one above all the others. For Organizers, the *Market* and *Opinion* worlds.

Other work established a relationship between the *Market* and *Industrial* worlds with the conventional food system [46]. In the same way, other authors have established that the economic aspects are the ones that determine the participation of consumers and producers in these initiatives [11]. However, it is suggested that, beyond the differences that can be observed, a series of common principles are presented in these spaces: promotion of sustainable livelihoods based on proximity, solidarity, trust and collectivism (the *Civic* world) [47]. To this respect, other work mentioned that farmers with a strong commitment to local food markets gave more importance to the civic commitment. The same does not apply for those that participate in multiple types of food supply chains [86].

Furthermore, a study in Prague [25] on the rapid growth of producers' markets, detected that these spaces have been co-opted by organizers of the conventional agri-food system. Although there is an interest in supporting producers' families, which would indicate a commitment to the *Civic* world, they prioritize the quality of the product over the personal relationships (in a clear tendency towards the Market world). That is the reason they do not represent a social movement, but a niche of heterogeneous organizers with compatible goals.

One of the most criticized aspects of these spaces is that although they manage civic stances, they are classified as exclusive places for consumers who can pay high prices for healthy foods. There is an interest also in promoting equitable access to these products [87], so as more consumers and producers get involved, prices will tend to decrease [88].

Some works defined the conventions as a system of reciprocal expectations about the behavior of others, a product of strategic negotiations among participants with knowledge and, to some degree, complicity in the structures they have created [51,53]. This is why there are three dimensions: (a) rules of spontaneous individual actions, (b) construction of agreements between people, and (c) institutions. They are practices, routines, agreements [89] and their associated informal and institutional forms, which link the different acts through mutual expectations (the "conventions") [53].

From the obtained results, it can be affirmed that different conventions can become Consumers' solutions and decisions to influence the management and practices of producers. They can also materialize into the different strategies developed to improve the features, attributes and characteristics of their products, which correspond to or are an answer to the different common principles of perceived justification [73], as these initiatives are constantly changing and improving in order to reach more consumers.

It is assumed that any form of coordination in the economic, political and social life (like the one present in chains and networks) requires an agreement that implies the construction of common perceptions about the structural context. These perceptions are reference points that the actors accumulate in their interactions [53,54].

## 6. Conclusions

The theoretical contributions of the article establish that CT allows for the analysis of reciprocal expectations about the behavior of others [57]. The obtained results confirm the usefulness of the applied approach to address the perceptions of Organizers, Producers and Consumers in the ecological markets in the south of Spain. Therefore, Hypothesis 1 and 3 were confirmed.

It is shown that the economic aspects linked to individual interests or products are not the only ones that form this activity. There is also a series of social factors related to the collective good and this is a characteristic of the commercialization short channels.

The three types of actors showed more interest in how these activities benefit the environment, those participating, the equality relationships, the respect and the opportunity for providing more access to these products (*Civic* world). The development of trust relationships based on the direct contact among the members of the markets and family participation was also important. This development stands out in the short food commercialization chains of the alternative markets (*Domestic* world), specially for Producers. Besides this, the three actors agreed about the following: the perceptions related to the reputation of the products, the importance of promoting these markets, the use of a trademark or certification to support quality (*Opinion* world). Lastly, they also agreed on the emotions that can be present when visiting these initiatives (*Inspired* world). Therefore Hypothesis 2 was confirmed.

Despite showing common perceptions, differences were also found. This is attributed to the role played by each actor and to socioeconomic aspects (gender, age, educational level, marital status). The most outstanding differences were found in the perceptions related to the interest about the adequate operation of markets for selling quality foods guaranteeing an extra price (*Market* world). These aspects were more important for Organizers and Producers. The latter showed a commitment with their work, the control of the production and the selling, and an adequate organization of the markets, considering that the amount of products is sufficient for the demand.

Results confirmed the importance of knowing what is happening in societies from the point of view of food perceptions. In this sense, it can be observed that positive perceptions are not the only ones present in these markets. There are also tense moments among participants, and interaction and negotiation allow for agreements to be reached, which enables the adequate progress of the activities). In this way, the behavior dynamics of the actors participating in these activities begins to be understood, being one of the first works of this type developed in the south of Spain in ecological markets.

About managerial implications, results can materialize into different strategies to improve these initiatives and reach more consumers.

## 7. Limitations and Future Research Directions

It is recognized that due to the exploratory nature of the study, the work has methodological limitations. One of those is the type of sampling: the study only included those willing to participate, so an unintended bias may be present. Another limitation is that the government actors' were not considered; future research should incorporate their perspective as their perception is essential to implement food policies to develop AFNs and increase consumer purchases at short chains.

**Author Contributions:** S.Y.E.-L.: design of research, data collection, analyses of information, interpretation of results, writing of manuscript and corrections. A.E.-O.: design of research, analyses of data, interpretation of results, writing of manuscript and corrections. S.A.-C.: interpretation of results, theoretical support and revision of manuscript. All authors have read and agreed to the published version of the manuscript.

**Funding:** Consejo Nacional de Ciencia y Tecnología (CONACYT) provided a doctorate grant to Sttefanie Yenitza Escobar-López and the international grant for research in the University of Cádiz.

**Institutional Review Board Statement:** Research procedures with interviewees followed methods accepted by Universidad Autónoma del Estado de México and Universidad de Cádiz.

**Informed Consent Statement:** All persons interviewed knew about the objectives of the research, there identity in not disclosed, and agreed to respond to interviews

**Data Availability Statement:** Data is available from the correspondent authors upon reasonable request.

**Acknowledgments:** Thanks to Consejo Nacional de Ciencia y Tecnología (CONACYT) for the support offered to Sttefanie Yenitza Escobar-López with the doctorate scholarship and for the international scholarship for research in the University of Cádiz.

**Conflicts of Interest:** The authors declare no conflict of interest.

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
