# Peer review of "Alternative Food Networks: Perceptions in Short Food Supply Chains in Spain"

_sustainability, doi:10.3390/su13052578_

Round 1

Reviewer 1 Report

Dear Authors,

My comments and suggestions are as follows:

1. It is suggested to rewrite whole abstract with proper novelty and findings of your research. Focus this more on your results than on the problem stating. Add information about the methodology.

2. The introduction is too long. Try to shorten the introduction and better justify the research gaps. Delete unnecessary paragraphs, e.g. the first paragraph seems redundant.

3 The article does not have a "literature review" section! To be more specific, what has been done by scholars should be well categorized first. Further, what has not bee done yet or what inconsistent findings still exist which need to be further addressed.

The literature review should be broader. It is worth to refer to the works of the following authors:

- Zimon, D.; Madzik, P.; Domingues, P. Development of Key Processes along the Supply Chain by Implementing the ISO 22000 Standard. Sustainability 2020, 12, 6176.
- Principato, L.; Ruini, L.; Guidi, M.; Secondi, L. Adopting the circular economy approach on food loss andwaste: The case of Italian pasta production.Res. Conserv. Recycl.2019,144, 82–89.
- Principato, L., Secondi, L., Cicatiello, C., & Mattia, G. (2020). Caring more about food: The unexpected positive effect of the Covid-19 lockdown on household food management and waste. Socio-Economic Planning Sciences, 100953.
. Lysenko-Ryba, K.; Zimon, D. Customer Behavioral Reactions to Negative Experiences during the Product Return. Sustainability 2021, 13, 448.
Wei, A.-P.; Peng, C.-L.; Huang, H.-C.; Yeh, S.-P. Effects of Corporate Social Responsibility on Firm Performance: Does Customer Satisfaction Matter? Sustainability 2020, 12, 7545.
etc.

4. It is worth develop research hypotheses. Hypotheses represent the key ideas that you are examining in order to answer your research questions. They need to be developed and argued in the literature review with support from the literature which demonstrates that they are valuable but not yet identified relationships.

5. The discussion section is very good.

6. Methods: Methodology is appropriate for the study. However, some of the statements required more clarifications. Please discuss in detail how you analyzed the data in order to present the key findings.

I hope the authors can work on these suggestions.
All the best!

Author Response

Reviewer 1

My comments and suggestions are as follows:

  1. It is suggested to rewrite whole abstract with proper novelty and findings of your research. Focus this more on your results than on the problem stating. Add information about the methodology.

Response:Correction was done as suggested.

Abstract:Alternative spaces for the consumption of non-conventional foods as short commercialization chains have been increased worldwide, also the interest in understanding the dynamics developed in those initiatives from a social approach has increased. This work aimed to analyze the Organizers, Producers, and Consumers' perceptions that participate in short food supply chains in ecological markets in the south of Spain. The Conventions Theory framework was used by applying a quantitative and qualitative methodology. A total of 159 questionnaires were applied (three to Organizers, 15 to Producers and 177 to Consumers). The questionnaire considered items related to social conventions for seven worlds (Domestic, Civic, Market, Industrial, Opinion and Inspired) and sociodemographic information. Kruskal-Wallis and Mann-Whitney tests were used to analyze the information. Results show that coincidences and divergences are observed in the importance given to the worlds; the Civic, Inspired and Opinion worlds are perceived similarly by the three types of actors and are related to the interest in how these activities benefit the environment. Differences are attributed to the role played by each type of actor and to socioeconomic aspects. Results can materialize into different strategies to improve these initiatives and reach more consumers.

  1. The introduction is too long. Try to shorten the introduction and better justify the research gaps. Delete unnecessary paragraphs, e.g. the first paragraph seems redundant.

Response: Thank you for the suggestion, the manuscript was revised for greater clarity and coherence.  Research gaps were highlighted.

Authors decided to leave the first paragraph as it stands since it gives a context according to the journal.

3 The article does not have a "literature review" section! To be more specific, what has been done by scholars should be well categorized first. Further, what has not bee done yet or what inconsistent findings still exist which need to be further addressed.

Response:Thank you for the suggestion. The editor sent a template with the structure that we should follow that did not include literature review. Nevertheless, the manuscript was revised for greater clarity and coherence and re-structured according to the reviewer´s suggestion, including the literature review. The following information was added:

The issue of consumer or customer satisfaction is a relevant and current research topic at all commercial levels, as has been shown for large firms where high customer satisfaction can diminish the negative effects of corporate social irresponsibility [18, 19]. In this sense, standard regulations along food supply chains may positively improve each stage of the supply chain [20].

Although the organic sector is still relatively small, the demand for organic food is increasing around the world [29]….

These urban spaces have slowly become consolidated as places for the responsible sale and consumption of food [38,39], that provide three types of benefits: on health, well-being through the solidarity to farmers, and in minimizing the ecological footprint of production [40].

However, the main gap of those studies is that production, consumption and retail channels [16,29,45-49] and organizers [25] have been analyzed separately.

The lack of studies may be due to the absence of methodological tools that allow analyzing the various actors involved under similar bases, regardless of their role in the AFNs. 

Conventions are defined as a system of reciprocal expectations in the behavior of others [54], that may be analyzed [58] since these are centered in scoring, judging, justifying and criticizing forms on how links are established between cognitive, moral and material concerns [43].

In that sense, people are assessed through their action [53,59] in different worlds of justification, stating that actions are legitimized by support on particular view points of the common good [58].

Therefore, TC enables the better understanding what regulates actions from different actors and how they perceive their actions [56], that is, ways of valorization to face criticism and justification [58].

There is a gap in the knowledge on a methodological approach that considers all of the different actors involved in AFNs. This is why the CT approach can contribute to their knowledge.

The literature review should be broader. It is worth to refer to the works of the following authors:

- Zimon, D.; Madzik, P.; Domingues, P. Development of Key Processes along the Supply Chain by Implementing the ISO 22000 Standard. Sustainability 2020, 12, 6176.
- Principato, L.; Ruini, L.; Guidi, M.; Secondi, L. Adopting the circular economy approach on food loss andwaste: The case of Italian pasta production.Res. Conserv. Recycl.2019,144, 82–89.
- Principato, L., Secondi, L., Cicatiello, C., & Mattia, G. (2020). Caring more about food: The unexpected positive effect of the Covid-19 lockdown on household food management and waste. Socio-Economic Planning Sciences, 100953.
-Lysenko-Ryba, K.; Zimon, D. Customer Behavioral Reactions to Negative Experiences during the Product Return. Sustainability 2021, 13, 448.
-Wei, A.-P.; Peng, C.-L.; Huang, H.-C.; Yeh, S.-P. Effects of Corporate Social Responsibility on Firm Performance: Does Customer Satisfaction Matter? Sustainability 2020, 12, 7545.

Response: The author decided not to include the following suggested references, since after thoroughly revised, we considered they are not related to the objective of this work.

- Principato, L.; Ruini, L.; Guidi, M.; Secondi, L. Adopting the circular economy approach on food loss andwaste: The case of Italian pasta production.Res. Conserv. Recycl.2019,144, 82–89.
- Principato, L., Secondi, L., Cicatiello, C., & Mattia, G. (2020). Caring more about food: The unexpected positive effect of the Covid-19 lockdown on household food management and waste. Socio-Economic Planning Sciences, 100953

On the other hand,the following references were added:

  1. Field, A. Discovering statistics using SPSS (4th ed.). Sage Publications. Great Britain 2013.
  2. Lysenko-Ryba, K.; Zimon, D. Customer behavioral reactions to negative experience during the product return. Sustainability2021, 13, 448; https://doi.org/103390/su13020448.
  3. Renard, M. Fair trade: quality, market and conventions. Rural Stud.2003, 19, 87–96.
  4. Rodríguez-Bermúdez, R.; Miranda, M.; Orjales, I.; Ginzo-Villamayor, M.J.; Al-Saufi, W.; López-Alonso, M. Consumers´ Perception of and Attitude Towards Organic Food in Galicia (Northern Spain). J. Consum. Stud2019, 44, 206-219; https://doi.org/10.1111/ijcs.12557.
  5. Rojas-Rivas, E.; Espinoza-Ortega, A.; Martínez-García, C.G.; Moctezuma-Pérez, S.; Thomé-Ortiz, H. Exploring the perception of Mexican urban consumers toward functional foods using the Free Word Association technique. of Sens. Stud.2018. e12439;https://doi.org/10.1111/joss.12439.
  6. Roldán, H.N.; Gracia, M.A.; Santana, M.E.; Horbath, J.E. Los mercados orgánicos en México como escenarios de construcción social de alternativas. POLIS Rev. Latin.2016, 15, 43, 1–18. https://doi.org/10.4067/S0718-65682016000100027.
  7. Neulinger, A.; Bársony, F.; Gjorevska, N.; Lazányu, O., Pataki, G.; Takács, S.; Török, A. Engagement and subjective well-being in alternative food networks: The case of Hungary. J. Consum. Stud. 2020, 44, 306-315; https://doi.org/10.1111/ijcs.12566.
  8. Nigh, R.; González, A.A. Reflexive Consumer Markets as Opportunities for New Peasant Farmers in Mexico and France: Constructing Food Sovereignty Through Alternative Food Networks. and Sust. Fd. Systms.2015, 39, 3, 317–341; https://doi.org/10.1080/21683565.2014.973545.
  9. Wei, A.P.; Peng, C.L.; Huang, H.C.; Yeh, S.P. Effect of corporate social responsibility on firm performance: Does customer satisfaction matter? Sustainability2020, 12, 7545; https://doi.org/10.3390/su12187545.
  10. Zimon, D.; Madzik, P.; Domingues, P. Development of key processes along the supply chain by implementing the ISO 22000 standards. Sustainability2020, 12, 6176; https://doi.org/103390/su12156176.

  1. It is worth develop research hypotheses. Hypotheses represent the key ideas that you are examining in order to answer your research questions. They need to be developed and argued in the literature review with support from the literature which demonstrates that they are valuable but not yet identified relationships.

 Response: Thank you for the suggestion. The next hypotheses were included:

Hypothesis 1. The global perspective of AFNs can be determined if the perceptions of all the types of actors that comprise them are analyzed.

Hypothesis 2. The CT is a useful approach to analyze the perception that the different actors involved on AFNs, have about these spaces.

Hypothesis 3. There are coincidences of objectives and perceptions of the different types of actors involved in the AFNs.

  1. The discussion section is very good.

Response:Authors appreciate the reviewer’s comments

  1. Methods: Methodology is appropriate for the study. However, some of the statements required more clarifications. Please discuss in detail how you analyzed the data in order to present the key findings.

Response:The methodology section has been thoroughly revised and rewritten as follows:

Non-probabilistic convenience sampling was used to select participants following the criteria that people were older than 18 and were interested and willing to participate [65]. A total of 159 persons participated (three Organizers, 15 Producers and 177 Consumers).

The methodology applied in this research was qualitative and quantitative, with semi-structured interviews and questionnaires. The questionnaire had two sections: 1) A series of items related to social conventions for each world (Domestic, Civic, Market, Industrial, Opinion and Inspired), items that were adapted according to multiple works [46,51,52,56,58,66] (Table 1) that were rated with a 5-point Likert scale (1=Total disagreement and 5=Total agreement) and 2) sociodemographic information (age, gender, family income, educational level and marital status).

Analysis of the information

A database was created in Excel and worked independently by type of actor (producer/organizer/consumer). The scores for the conventions of each world were added and averaged for each participant. These values were used to calculate the median and interquartile range for each world since Likert scales are non-parametric data [67]. The medians and interquartile range were used in the nonparametric Kruskal-Wallis and U-Mann-Whitney tests [67], to identify significant statistical differences (p <0.05) in the perception of the worlds per type of actor (Table 2). An amoeba graph was made to have a visual comparison of the worlds' values per type of actor (Figure 1).

Sociodemographic information was analyzed with descriptive statistics. The age variable was classified into sociological generations: Millennials (1983 to 2002), Generation X (1965 to 1983), or Baby boomers (1943 to 1964) [68].

Finally, the statements of 20 interviewed people were used to complement the analyzed information about the study's social practices [69].

Reviewer 2 Report

Thank you for submitting your work to Sustainability. The manuscript has definitely merit yet it needs to be significantly improved before publication. Please see my comments below:

  1. The introduction section is too long and there is no literature review section. It will be better to shorten the introduction section and add a literature review section. Although the theoretical underpinning was enabled with Conventions Theory, further information needs to be provided.
  2. The research gap must be clearly provided.
  3. The theoretical and practical contribution/implication section must be included.

Author Response

Reviewer 2

Thank you for submitting your work to Sustainability. The manuscript has definitely merit yet it needs to be significantly improved before publication. Please see my comments below:

Response:Authors appreciate the reviewer’s comments.

  1. The introduction section is too long and there is no literature review section. It will be better to shorten the introduction section and add a literature review section. Although the theoretical underpinning was enabled with Conventions Theory, further information needs to be provided.

Response: Thank you for the suggestion. The editor sent a template with the structure that we should follow that did not include literature review. Nevertheless, the manuscript was revised for greater clarity and coherence and re-structured according to the reviewed suggestion including the literature review and the theoretical basis.  The following information was added:

Conventions are defined as a system of reciprocal expectations in the behavior of others [54], that may be analyzed [58] since these are centered in scoring, judging, justifying and criticizing forms on how links are established between cognitive, moral and material concerns [43].

In that sense, people are assessed through their action [53,59] in different worlds of justification, stating that actions are legitimized by support on particular view points of the common good [58].

Therefore, TC enables the better understanding what regulates actions from different actors and how they perceive their actions [56], that is, ways of valorization to face criticism and justification [58].

  1. The research gap must be clearly provided.

Response:Research gaps were highlighted and rewritten as follows:

In the European context, there are different works which have dealt with AFNs. Some authors who have analyzed these markets determined that their development and growth depend to a large extent on the knowledge of the role of consumers and also on the satisfaction of their needs [29,41-43]. Although the potential importance of individual consumers and communities is noteworthy [44], the consideration of the rest of actors involved in food chains (in order to correctly understand their signifiers and practices) becomes of relevance. However, the main gap of those studies is that production, consumption and retail channels [16,29,45-49] and organizers [25] have been analyzed separately. Therefore, to obtain a global vision of AFNs, behaviors, motivations and perceptions of all involved actors should be taken into account [4]. In this manner, a solid alternative can be constructed to face the predominant ways for the distribution of foods [50] and to strengthen these proposals.

There is a gap in the knowledge on a methodological approach that considers all of the different actors involved in AFNs. This is why the CT approach can contribute to their knowledge. In that sense:

  1. The theoretical and practical contribution/implication section must be included.

Response:The theoretical contribution and practical implications were highlighted as follows:

The theoretical contributions of the article establish that CT allows for the analysis of reciprocal expectations about the behavior of others [58]. The obtained results confirm the usefulness of the applied approach to address the perceptions of Organizers, Producers and Consumers in the ecological markets in the south of Spain.

About managerial implications, results can materialize into different strategies to improve these initiatives and reach more consumers.

Limitations and future research directions

It is recognized that due to the exploratory nature of the study, the work has methodological limitations. One of those is the type of sampling; the study only included those willing to participate so that an unintended bias may be present. Another limitation is that the government actors' were not considered; then future research should incorporate their perspective; their perception is essential to implement food policies to develop AFNs. and increase consumer purchases at short chains.

Reviewer 3 Report

In the assessment of the paper submitted for the review, I specifically focused on the discussed issues, applied methodology, the substantive content of the paper and its structure.

The considerations conducted in the paper are focused on such categories as: alternative food networks, short chains, ecological markets, nutritional trends.

The subject area discussed in the paper should be considered interesting.

The value of the paper results from appropriate combination of literature studies with the results of an empirical research.

Although the topic of this research study is interesting, I think this paper should apply the comments indicated below to increase the quality of contributions and findings.

To improve the quality of the paper I would suggest to:

- develop the description of methodology and criteria of respondents’ selection; expand the information about the applied research procedure,

- explain the choice of the statistical tests,

- add the research questions or theses,

- indicate the theoretical contributions of the article,

- develop the description of the limitations of research,

- indicate the directions for further research,

- expand the managerial implications in the article.

Author Response

Reviewer 3

In the assessment of the paper submitted for the review, I specifically focused on the discussed issues, applied methodology, the substantive content of the paper and its structure.

The considerations conducted in the paper are focused on such categories as: alternative food networks, short chains, ecological markets, nutritional trends.

The subject area discussed in the paper should be considered interesting.

The value of the paper results from appropriate combination of literature studies with the results of an empirical research.

Although the topic of this research study is interesting, I think this paper should apply the comments indicated below to increase the quality of contributions and findings.

Response: Authors appreciate the reviewer’s comments.

To improve the quality of the paper I would suggest to:

- develop the description of methodology and criteria of respondents’ selection; expand the information about the applied research procedure,

- explain the choice of the statistical tests,

Response:The methodology section has been thoroughly revised and rewritten as follows:

Non-probabilistic convenience sampling was used to select participants following the criteria that people were older than 18 and were interested and willing to participate [65]. A total of 159 persons participated (three Organizers, 15 Producers and 177 Consumers).

The methodology applied in this research was qualitative and quantitative, with semi-structured interviews and questionnaires. The questionnaire had two sections: 1) A series of items related to social conventions for each world (Domestic, Civic, Market, Industrial, Opinion and Inspired), items that were adapted according to multiple works [46,51,52,56,58,66] (Table 1) that were rated with a 5-point Likert scale (1=Total disagreement and 5=Total agreement) and 2) sociodemographic information (age, gender, family income, educational level and marital status).

Analysis of the information

A database was created in Excel and worked independently by type of actor (producer/organizer/consumer). The scores for the conventions of each world were added and averaged for each participant. These values were used to calculate the median and interquartile range for each world since Likert scales are non-parametric data [67]. The medians and interquartile range were used in the nonparametric Kruskal-Wallis and U-Mann-Whitney tests [67], to identify significant statistical differences (p <0.05) in the perception of the worlds per type of actor (Table 2). An amoeba graph was made to have a visual comparison of the worlds' values per type of actor (Figure 1).

Sociodemographic information was analyzed with descriptive statistics. The age variable was classified into sociological generations: Millennials (1983 to 2002), Generation X (1965 to 1983), or Baby boomers (1943 to 1964) [68].

Finally, the statements of 20 interviewed people were used to complement the analyzed information about the study's social practices [69].

- add the research questions or theses,

Response: Thank you for the suggestion. The next hypotheses were included:

Hypothesis 1. The global perspective of AFNs can be determined if the perceptions of all the types of actors that comprise them are analyzed.

Hypothesis 2. The CT is a useful approach to analyze the perception that the different actors involved on AFNs, have about these spaces.

Hypothesis 3. There are coincidences of objectives and perceptions of the different types of actors involved in the AFNs.

- indicate the theoretical contributions of the article,

Response:the theoretical contribution was highlighted as follows:

The theoretical contributions of the article establish that CT allows for the analysis of reciprocal expectations about the behavior of others [58]. The obtained results confirm the usefulness of the applied approach to address the perceptions of Organizers, Producers and Consumers in the ecological markets in the south of Spain.

- develop the description of the limitations of research,

- indicate the directions for further research

Response:Limitations and future research directions were incorporated.

Limitations and future research directions

It is recognized that due to the exploratory nature of the study, the work has methodological limitations. One of those is the type of sampling; the study only included those willing to participate so that an unintended bias may be present. Another limitation is that the government actors' were not considered; then future research should incorporate their perspective; their perception is essential to implement food policies to develop AFNs. and increase consumer purchases at short chains.

- expand the managerial implications in the article.

Response:About managerial implications, results can materialize into different strategies to improve these initiatives and reach more consumers.

Round 2

Reviewer 1 Report

Dear authors, thank you for your answers and explanations.

I only have one point. There is no need to create a separate section for "Limitations and future research directions". I propose to include it in section 6. 

I encourage you to continue researching in this field.

Reviewer 3 Report

Thank You for introducing my suggestions in article.